# Analysis of the Ability to Tolerate Body Balance Disturbance in Relation to Selected Changes in the Sagittal Plane of the Spine in Early School-Age Children

**DOI:** 10.3390/jcm11061653

**Published:** 2022-03-16

**Authors:** Piotr Kurzeja, Bartłomiej Gąsienica-Walczak, Katarzyna Ogrodzka-Ciechanowicz, Jarosław Prusak

**Affiliations:** 1Institute of Health Sciences, Podhale State College of Applied Sciences, 34-400 Nowy Targ, Poland; piotrkurzeja@op.pl (P.K.); bgw01@interia.pl (B.G.-W.); j.prusak2020@gmail.com (J.P.); 2Institute of Clinical Rehabilitation, Faculty of Motor Rehabilitation, University of Physical Education, 31-571 Krakow, Poland; 3Institute for Tuberculosis and Lung Diseases, 34-700 Rabka-Zdrój, Poland

**Keywords:** idiopathic scoliosis, postural balance, posture

## Abstract

The study aimed to estimate the ability to tolerate body balance disturbance in relation to selected changes in the sagittal plane of the spine in early school-age children. The study involved 189 children with an average age of 8.3 ± 0.7 years (aged 7–10). The tests included an interview, clinical examination (measurement of body weight and height, assessment of the course of the spinous processes of the thoracic and lumbar vertebrae, assessment of the location of selected anatomical landmarks of the torso), and a physical examination in which the shape of the spine surface was examined with the use of the photogrammetric method and the moiré effect projection. Body balance disturbance tolerance skills (BBDTS) were measured with the rotational test (RT). In the rotational test, the results of body balance disturbance tolerance skills show a slight but statistically significant correlation with the bodyweight of the examined children (Rs = 0.35, *p* < 0.001). This relationship was also statistically significant in the groups by gender. Among the measured indicators of the curvature of the spine in the sagittal plane, the correlation with the RT test result was mostly related to the α angle and the value was Rs = 0.15 (*p* = 0.04). In the group of girls, this correlation was stronger and amounted to Rs = 0.26 (*p* = 0.015). Among other measured correlations, the dependence of variables such as the bodyweight of the subjects and the α angle was shown. In conclusion, increasing lumbar lordosis results in the deterioration of balance disturbance tolerance skills. As body weight increases, body balance disturbance tolerance skills decrease.

## 1. Introduction

Postural defects are one of the most common health problems of modern society. The percentage of people with abnormal body posture is increasing. It is estimated that 34–50% of children and adolescents have postural defects of various degrees of severity [1]. The combination of the nervous system and the musculoskeletal system is responsible for the proper functioning of the human body [2]. The central nervous system develops most rapidly in the first years of a child’s life, to reach its full functionality in adolescence [3]. 

So far, there is no clear answer in the literature about the relationship between postural stability and body posture itself. Walaszek et al. [4] suggest that there is a relationship between postural control and body posture indices, but only in the scope of selected parameters and differentiating between the sexes. The said authors observed that in the group of girls, the position of shoulder blades was statistically significantly correlated with the global Y-balance test (YBT) result for the right lower extremity. In the group of boys, the position of waist triangles was statistically significantly correlated with global YBT results for the right and left lower extremities [4].

It is also recognized that due to posture defects, a child’s physical fitness decreases, which in turn leads to a reduction in the control of body balance and a weaker response to a sudden loss of balance [5]. On the other hand, Ostrowska and Skolimowski [6] proved that people with scoliosis show greater imbalance and need more time to regain a stable posture. 

Selected disorders of postural stability in children may therefore have consequences in adulthood, hence the need to diagnose the body posture of children and adolescents is justified, and also in terms of existing therapeutic programs. In physiotherapeutic practice, particular importance should be given to research not only on body posture but also on postural stability. Increasing the effectiveness of the therapeutic process, including exercise, also depends on identifying the relationship between body posture and postural stability.

The alignment in the sagittal plane is the most difficult criterion for describing correct body posture, while both the frontal and transverse planes are usually considered symmetrical [7,8].

Physiological curvatures of the spine in the sagittal plane are a typical feature of good body posture. The cervical and lumbar spine is curved forward (lordosis), while the thoracic spine is curved back (kyphosis). The head remains horizontal, i.e., eyes are on a horizontal plane, while the chin is just above the breastbone. The pelvis is tilted forward and the joints of the lower extremities remain in a neutral position [9,10].

In 25–60% of children and adolescents, body posture disorders occur in the form of a rounded back, sloped shoulders or excessive pelvic tilt [11,12,13,14].

The scarce information on the relationship between postural control and posture defects, especially in the sagittal plane, seems important for analysing this topic.

The study aimed to estimate the ability to tolerate body balance disturbances in relation to selected changes in the sagittal plane of the thoracic and lumbar spine in early school age children.

## 2. Materials and Methods

This observational (cross-sectional) study was conducted in compliance with the Strengthening the Reporting of Observational Studies in Epidemiology (STROBE) Statement: guidelines for reporting observational studies [15].

The research was carried out in five primary schools from the Śląskie Voivodeship in the Psary Commune. The research was conducted from November to December 2019.

The first author enrolled students for the study from five primary schools in the Śląskie Voivodeship in the Psary Commune.

Inclusion criteria:age 7–10 years,no health issues that might affect the test result, i.e., diagnosed neurological diseases, posture defects, musculoskeletal injuries (vision defects, disturbances in neuromuscular coordination, excessive body weight, past injuries of the spine and lower limbs),written consent of a parent (guardian) to participate in the study,consent of school principals to the examination,attending primary schools in the Psary commune,the ability to communicate with the examined child necessary to conduct the examination (the child’s willingness to undress and prepare for the examination).

Exclusion criteria:no written consent of the child’s parents or guardians,age of the examined child under 7 and over 10 years of age,diseases that make it impossible to conduct the test or that may affect its result (injuries, fractures of the lower or upper limbs, or spine),inability to communicate with the examined child necessary for the examination (the child’s refusal to undress and prepare for the examination).

The research consisted of four parts:Interview, including information on the date of birth, age in years and months.The clinical trial included:
bodyweight measurement,measuring the body height in an upright position,assessment of the alignment of the spinous processes of the thoracic and lumbar vertebrae,assessment of the location of selected anatomical landmarks of the torso: processes of the scapulae and inferior angles of the scapulae, triangles of the waist, anterior superior and posterior superior iliac spines, and the greater trochanter.

Weight and height were measured with a verified medical column scale C315.60/150.OW-3—a 100–200 cm height measuring device (UNIWAG—Professional electronic scales, Krakow, Poland). A medical skin marker from Covidien was used to mark the characteristic anthropometric points on the skin (Medtronic, Minneapolis, MN, USA).

3.Physical examination in which the shape of the ridge surface was analysed with the use of the photogrammetric method and the moiré effect projection (Device for Computer Posture Assessment of the Body of the 4th Generation MOIRE system, Wrocław, Poland) (Figure 1).4.Assessment of body balance disturbance tolerance skills (BBDTS).

### 2.1. Outcome Measures

The projection moiré system is a computer technique of body posture analysis that uses light beam refraction for obtaining isolines that connect points of the same height. Nowadays, this technique most often uses an optical raster, which is built into a projection device. The image that is obtained is received by the camera, then transferred to a computer that uses a special program for the analysis of posture to process detailed data. The final result of the set of three-dimensional coordinates is a map of the body surface (a contour image with so-called moiré stripes). Its appearance depends on the surface that is illuminated (in this case, the back) and the distance of the object from the camera.

In this work, the authors used the MORA 4th Generation system. This device combines the advantages of the MORA/ISIS system used for spatial analysis, and systems that are based on markers and are related to the analysis of movement or gait. The MORA 4th Generation system corresponds to the test performance standards related to the assumption that it should be quick and simple [16].

As the device uses two built-in cameras, an image of the entire human figure can be obtained (Figure 1). Thanks to this, errors related to the positioning of the figure, such as, for example, knee flexion, head rotation, etc., are mostly eliminated [16].

Body balance disturbance tolerance skills (BBDTS) were measured with the rotational test (RT) [17,18]. The RT consists of performing six 360° (full-turn) jumps (rotation to the right and left) and landing after each such jump, so that both feet re-establish contact with a line from which the test subject starts, with the test lasting approximately 12 s.

Testing a person should be preceded by a detailed explanation of the rules and the tested person should be allowed to perform two attempts at a full-turn jump in the following order: right; left. Instructions for performing the test and evaluation criteria: the examined person is standing with his or her legs apart (legs hip-width apart) on a hard mattress, so that the midfeet are on a line (line not thicker than 1 cm). On the command ‘ready’ given by the principal investigator of the research, the examined person bends his or her knees and at hearing the word ‘right’ he or she jumps full-turn to the right and, after landing, the command ‘correction’ given by the principal investigator is a call to correct the posture (if he or she has lost contact with two feet, he or she puts the feet back to the initial position); when the command is ‘left,’ he or she jumps full-turn to the left. Once again, the command ‘correction’ is a call to correct the posture. The next cycle begins with rotation to the right, etc. (each “Jump-Landing-Correction” cycle takes about 2 s). The principal investigator of the research starts a stopwatch while pronouncing “r” during the first command, i.e., ‘right’, and stops it during the pronunciation of “t” in the last command, i.e., “left”. Time should be documented with an accuracy of 0.01 s. The test duration is supplementary information.

The tested person should not be pressured into trying to complete the test as soon as possible. Attention should be paid to the need to precisely perform individual tasks while maintaining the principle of adjusting commands to the rhythm of full-turn jumps and the necessary corrections. The motor effects of the test are documented by the assistant.

Landing after a jump with both feet on the designated line and maintaining balance means no error (score is “0”), no contact of one foot with the line after landing is assessed as “1” (first-degree mistake), “2” means neither foot is in contact with the line after landing (2nd-degree mistake), and ”3” means leaning against the ground with a hand/both hands or a fall (3rd-degree mistake). The overall result is the sum of the six tasks (consecutive full-turn jumps) and includes 0 to 18 possible points. “0” indicates a very high ability to tolerate imbalances, while “18” means the exact opposite of that assessment. Criteria for individual assessment of body balance disturbance tolerance skills (BBDTS) determined by RT results: very high (0–1), high (2–3), average (4–9), low (10–12), very low (13–15), insufficient (16–18).

### 2.2. Intervention

During the examination, it was necessary to darken the room and turn off the artificial lighting for the time of recording the image. The examined person was dressed only in underwear and barefoot. Before starting the examination, selected anatomical landmarks were marked on the child’s skin with a washable skin marker. The child was in a standing position for examination, the left and right posterior superior iliac spines (PSIS) were at an equal distance from the apparatus (the pelvic twist angle was 0°). PSIS were palpated by the attending physiotherapist. In this position, a few to a dozen or so shots were recorded, from which a single shot meeting the condition of the correct position of the pelvis and reflecting the most frequently appearing posture of the patient was then selected. The examination of one child, including preparation, took about 5 min. The tests were always carried out by the same person.

Three indices describing changes in the sagittal plane of the spine were analysed:ALPHA angle [α]—the inclination of the lumbosacral segment. This is the angle between the vertical line and the line between S1 (the spinous process of the first sacral vertebra) and the LL (apex of lumbar lordosis).BETA angle [β]—the inclination of the thoracic-lumbar segment. This is the angle between the vertical line and the line between LL (apex of lumbar lordosis) and KP (apex of thoracic kyphosis).GAMMA angle [γ]—the slope of the upper thoracic segment. This is the angle between the vertical line and the inclusive line KP (apex of thoracic kyphosis) and C7 (the spinous process of the seventh cervical vertebra).

After the examination, a rotational test was performed (RT). The tests were always carried out by the same person. Each child made two attempts at a full-turn jump in the order: to the right; to the left.

### 2.3. Statistical Analysis

In the statistical analysis, the arithmetic mean, median and standard deviation of the measured indicators were calculated. The normality of the distribution was checked using the Kolmogorov–Smirnov test with the Lilliefors correction and the Shapiro–Wilk test. Group differences by gender were assessed using a test for single samples for two independent groups, Student’s *t*-test or Mann–Whitney U test in the absence of parametricity. To evaluate the differences between the groups according to the RT test result, the chi-square test was used. Differences were considered significant at *p* < 0.05. Depending on the normality of the distribution, Pearson’s or Spearman’s correlation was used for assessing the relationship between the indicators. All analyses were carried out using the STATISTICA 12.0 software package.

## 3. Results

Initially, the first author enrolled 217 children aged 7–10 for the research. Due to parent’s/guardian’s written resignation from the participation of a child in the study (*n* = 8), diseases that made it impossible to conduct the study (*n* = 11), or the lack of communication with the tested child necessary to conduct the study (*n* = 9), in the final stage of the study, 189 children were qualified. Table 1 presents detailed anthropometric data of the examined group. Figure 2 presents the qualification stage. The differences between the study groups were not statistically significant.

Table 2 presents the basic statistics of the values of the angles describing changes in the sagittal plane of the spine for the entire study group and by gender. The differences were not statistically significant.

The results of the RT test are presented in Table 3. The mean test result in the whole group was 6.5 ± 2.9 points, in the group of girls it was 6.2 ± 2.6, and in the group of boys, it was 6.7 ± 3.2. The difference in the groups by sex was not statistically significant. Taking into account the criteria of individual assessment of the level of body balance disturbance tolerance skills (BBDTS) determined by RT results, the results obtained in the test are presented in Figure 3 (differences for x^2^ = 72.7, *p* < 0.0001).

There was a slight but statistically significant correlation between RT test results and the bodyweight of the subjects (Rs = 0.35, *p* < 0.001) (Table 4, Figure 4). This correlation was also statistically significant considering the above and the division of the participants by gender (Table 5). Among the measured indicators of the curvature of the spine in the sagittal plane, the correlation with the RT test result was mostly correlated with the α angle, and the value was Rs = 0.15 (*p* = 0.04) (Table 4). In the group of girls, this correlation was even slightly stronger and amounted to Rs = 0.26 (*p* = 0.015) (Table 5, Figure 5). Other correlations were weaker and statistically insignificant (Table 6). Among other measured correlations, the dependence of variables such as the bodyweight of the subjects and the α angle was shown (Rs = 0.30, *p* < 0.001) (Table 4).

Figure 6 shows the results of the variable α in each group according to the RT test score. The differences were not statistically significant.

## 4. Discussion

The assessment of body posture control has been the subject of many publications [19,20,21]. The relationship and interactions between posture and movement have been an issue that has long been described in neurology. According to Ivanenko and Gurfinkel [22], the central nervous system (CNS) combines mobility with stability. The latter was best described by Sherrington over a hundred years ago: “posture follows movement like a shadow” [23]. Tonic muscle activity and postural control require specialised neural circuits. Adequate postural tension is an integral part of any movement, and disturbances in muscle tone can, in turn, affect performance. To understand posture and movement control, it is important to understand how the neuromuscular tone of posture and movement is generated and maintained [22]. The relationship between body imbalances and posture defects is still rarely described. The aim of the author’s own research was to analyse the changes that occur in the sagittal plane of the thoracic and lumbar spine in early school children and to determine the relationship with selected parameters related to maintaining body balance.

The rotational test (RT) was used in the research to assess body balance disturbance tolerance skills (BBDTS). It is a very useful tool for screening tests [18]. The results of the research presented in this article indicate, first of all, the relationship between the value of the α angle and the bodyweight of children, which may indicate that the greater the body weight, the higher the risk of changes in the sagittal plane. The research results indicate that the RT test result was worse as body weight increased. Additionally, the results indicate that with the increase of the α angle, the RT test result was also worse. This may mean that increasing lumbar lordosis results in the deterioration of the ability to tolerate body imbalances. The authors believe that in the next stages of similar research, it would also be worth assessing the angle of anterior pelvic tilt, the value of which may significantly affect the size of lumbar lordosis in children. It also seems important to assess possible muscle contractions that have an impact on this situation (e.g., for the iliopsoas muscle).

It can be assumed that selected postural disturbances are the result of neuromuscular imbalance [24]. Depending on the balance of muscles, individual body segments assume specific positions. It is assumed that weak abdominal and buttock muscles and weak contraction of hip flexors may lead to increased anterior pelvic tilt, which in turn results in lumbar hyperlordosis [25,26]. Such a situation may occur in disorders related to excessive body weight. The results of the correlation of body mass with the value of the α angle obtained in this study seem to confirm this relationship. Among all the measured spine curvature indices, only the value of the α angle correlated to the greatest extent with the results obtained in the RT test. Interestingly, this relationship was especially true for the group of girls, although the mean values of body weight in this group were lower than in the group of boys (nonsignificant difference), and the value of the α angle correlated to a slightly greater extent with the RT test result in the group of boys.

Some authors suggest that the correction of body posture (e.g., correction of a sway back) should take into consideration the control of body balance [27,28,29]. Current models of body stability control oversimplify the human body by treating it as an inverted pendulum in which the mass is concentrated at one point [30,31]. Although the inverted pendulum model reflects well the regulatory principles of the central nervous system (CNS), it significantly oversimplifies the reality. The torso is a very flexible system consisting of numerous joints, and by activating the muscles, the CNS can change the position of individual trunk segments in relation to each other, which automatically shifts the position of the centre of body mass (COM) and, as a result, shifts the centre of pressure (COP) [21,32]. Body posture defects, disturbing the position of the pelvis or individual segments of the spine in the three-dimensional space, may consequently lead to displacements of the centre of body mass (COM) [21]. Conscious control of body posture initially causes a reorientation, especially of the pelvis, as well as lumbar and thoracic segments. Thus, it activates the appropriate muscles to bring the individual joints (hip joint, shoulder joint, upper cervical spine joints) closer to a common vertical line. Conversely, deterioration in posture is characterised by the fact that these joints move apart, and the torso is rather passively stabilized by ligaments and joint capsules [20].

Research shows that the ability to maintain balance can be developed through the use of appropriate exercise. Research by Kalina et al. among young female athletes practising vaulting proves that the sports disciplines in which intense disturbances of balance occur very often significantly increases the tolerance of the system to such environmental stimuli [17].

Stability and postural regulation are related because the CNS must control stability when the position of body segments changes [32,33,34]. The literature describing the relationship between body posture and changes in stability is scarce. The relationships between body posture and parameters related to balance in various diseases have been investigated by Drzal-Grabiec et al. [35], Lopes et al. [36], and Willigenburg et al. [37]. In their studies on groups of adolescents with scoliosis, Nault et al. [38] and Stylianides et al. [39] found that imbalance in an upright position was associated with altered body posture parameters. On the other hand, Nagymáté et al. [40] found no significant deterioration in postural control in children with impaired posture. Massion [32] stated, however, that the process of correcting body posture is associated with both the control of balance and the stabilization of a particular position. The results of our own research indicate a correlation between changes in the sagittal plane of the lumbar spine and the control of body posture measured with the use of the RT test. Increasing lumbar lordosis results in the deterioration of body balance disturbance tolerance skills.

Our own research shows that stability control and posture control are interdependent mechanisms and that the complex interaction between them is not yet fully understood. According to Mrozkowiak [41], the total length of the spine, the angle of the torso, and the lateral deviation of the spine positively correlate with the height of the body, the angle of thoracic kyphosis and the depth of thoracic kyphosis. In turn, the angle of lumbar lordosis and the angle of pelvic tilt negatively correlate with the height of the body. In their own research, the authors of the presented study found no correlation between body height and RT results.

In the future, it would be worthwhile to broaden the analysis by including global postural reeducation (GPR) exercises in therapy. As reported in the GPR literature, it positively influences the improvement of function and the reduction of pain, and is an effective method for treating spinal disorders [42,43,44]. Studies with the use of stabilographic platforms are one of the most clinically important ways to control the progress of therapy, so their use will constitute the basis for the continuation of own research by the present authors.

### Study Limitation

This study is not without limitations. In order to unequivocally determine the relationship between selected parameters related to maintaining body balance and changes in the curvature of the spine in the sagittal plane in children, a test with eyes closed should also be performed.

There are no unambiguous reports in the literature confirming whether and how postural stability correlates with body posture. Some reports deny, and some confirm, the existence of mutual relations between body posture and balance. Therefore, it is important to continue research to unequivocally define the scope of the relationship between body balance and posture defects. Further research and results should contribute to the development of therapeutic programs based on targeted exercises for children with postural defects and scoliosis.

## 5. Conclusions

Increasing lumbar lordosis results in the deterioration of body balance disturbance tolerance skills.As body weight increases, body balance disturbance tolerance skills decrease.As body weight increases, there is a risk of changes in the anteroposterior curvature of the spine.

## Figures and Tables

**Figure 1 jcm-11-01653-f001:**
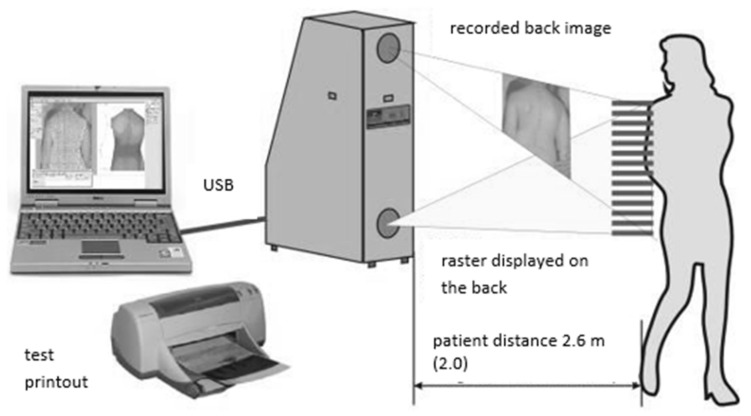
The moiré system [16].

**Figure 2 jcm-11-01653-f002:**
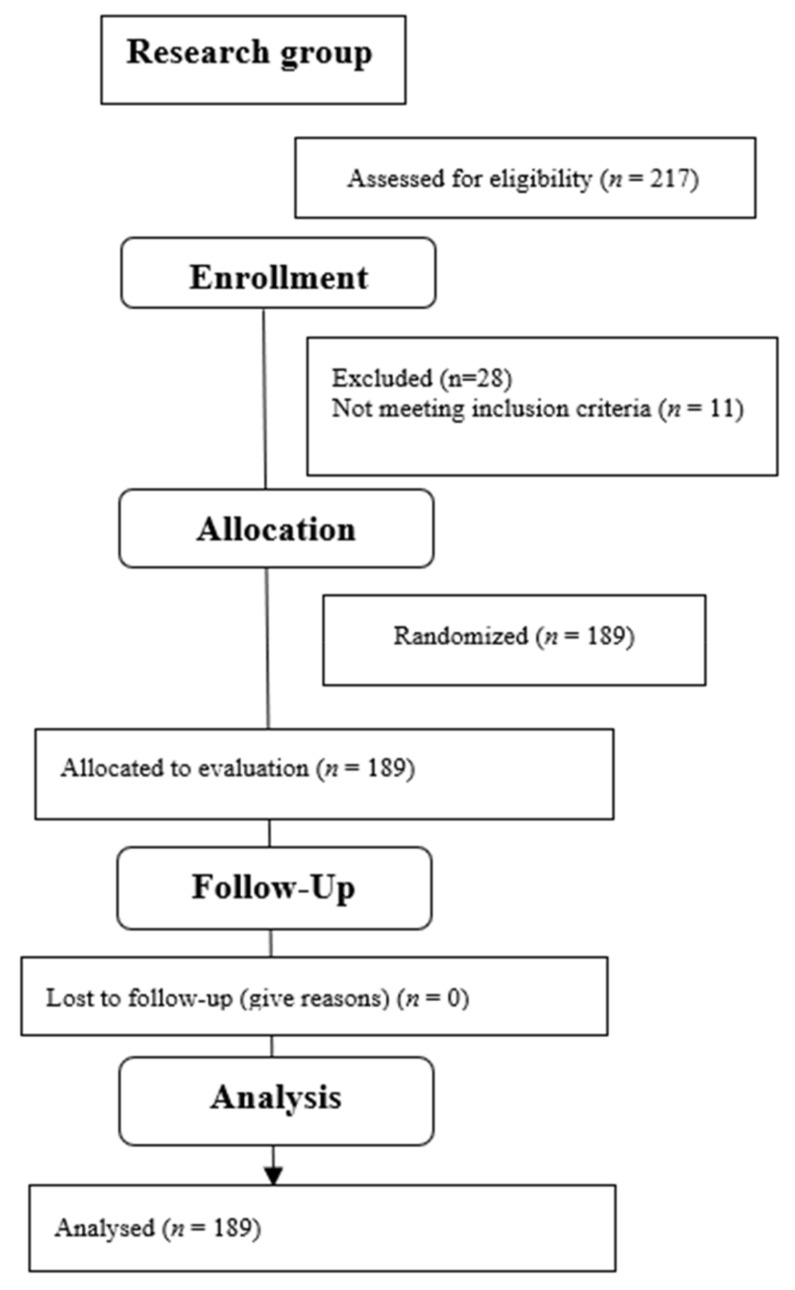
Flow diagram.

**Figure 3 jcm-11-01653-f003:**
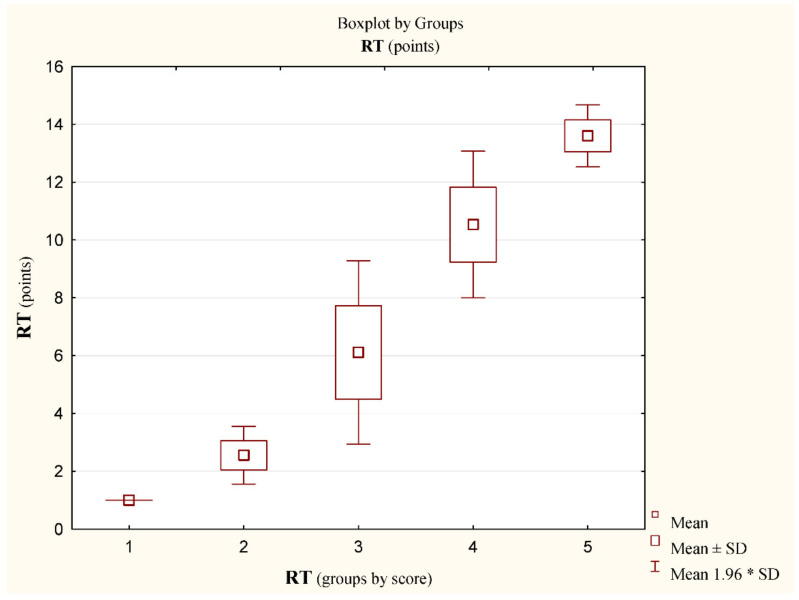
Results; score (pts) in the rotational test RT, (n) x ± SD; median. 1—Very high (2) 1.0 ± 0.0; 1.0; 2—High (27) 2.6 ± 0.5; 3.0; 3—Average (124) 6.1 ± 1.6; 6.0; 4—Low (31) 10.7 ± 0.8; 10.0; 5—Very Low (5) 13.6 ± 0.54; 14.0; 6—Insufficient (0). Differences between groups: x^2^ = 72.7 (*p* < 0.0001). * stands for the multiplication sign.

**Figure 4 jcm-11-01653-f004:**
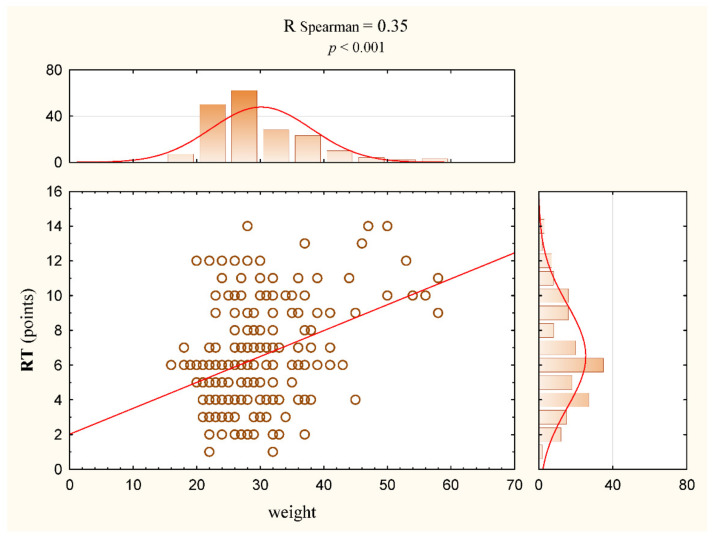
Correlation between weight and RT variables (circle) for the study group (all participants) with regression line (red line).

**Figure 5 jcm-11-01653-f005:**
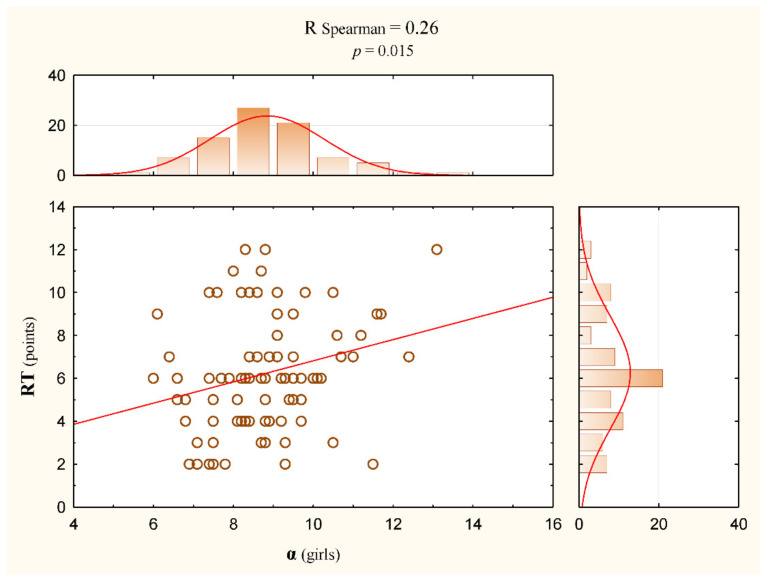
Correlation between α and RT variables (circle) in the group of girls with regression line (red line).

**Figure 6 jcm-11-01653-f006:**
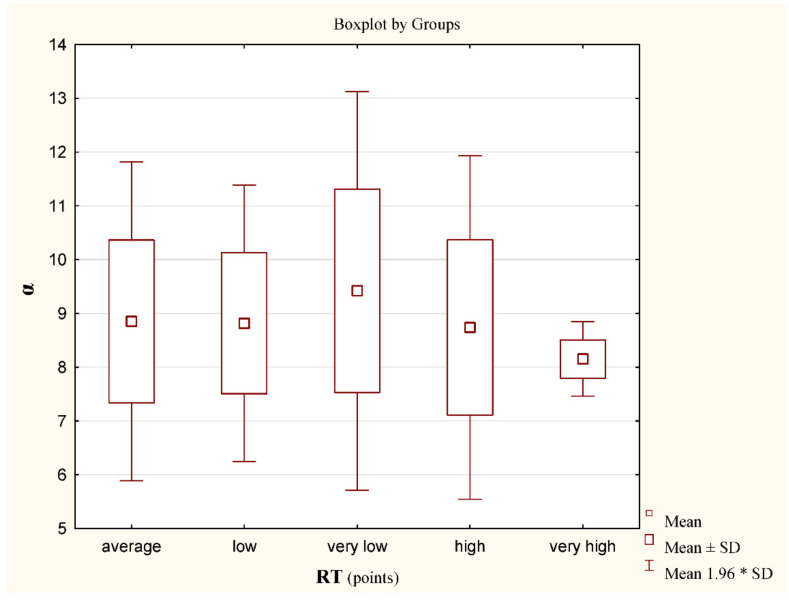
Results; mean values for the α angle for individual groups according to the scores in the rotational test RT, (n) x ± SD; M. Very high (2) 8.15 ± 0.4; 8.15; High (27) 8.7 ± 1.6; 8.6; Average (124) 8.8 ± 1.5; 8.8; Low (31) 8.8 ± 1.3; 8.9; Very Low (5) 9.4 ± 1.9; 9.5; Insufficient (0). Differences between groups x^2^ = 4.6 (*p* = 0.33). * stands for the multiplication sign.

**Table 1 jcm-11-01653-t001:** Study group.

Variable	All *n* = 189 x ± SD; M (Min/Max)	Girls *n* = 85 x ± SD; M (Min/Max)	Boys *n* = 104 x ± SD; M (Min/Max)
Age (years)	8.3 ± 0.7; 8.3 (7/9.8)	8.3 ± 0.7; 8.2 (7/9.8)	8.3 ± 0.7; 8.3 (7/9.8)
Height (cm)	131.3 ± 7.4; 131 (115/150)	130.3 ± 7.0; 130 (115/146)	132.1 ± 7.6; 133 (115/150)
Body weight (kg)	30.0 ± 7.9; 28.0 (16/58)	29.7 ± 7.9; 28.0 (16/58)	30.3 ± 7.9; 28.0 (18/58)

x—mean; SD—standard deviation; M—median; *n*—number of participants.

**Table 2 jcm-11-01653-t002:** Results describing changes in the sagittal plane of the spine in the study group.

Variable [°]	All *n* = 189 x ± SD; M	Girls *n* = 85 x ± SD; M	Boys *n* = 104 x ± SD; M
α	8.8 ± 1.5; 8.8	8.8 ± 1.4;8.8	8.8 ± 1.6; 8.9
β	7.9 ± 1.2; 8.0	8.0 ± 1.2; 7.9	7.8 ± 1.1; 8.0
γ	10.4 ± 1.5; 10.4	10.2 ± 1.6; 10.4	10.6 ± 1.3; 10.5
α + β	16.7 ± 1.8; 16.6	16.8 ± 1.7; 16.6	16.7 ± 1.9; 16.7
β + γ	18.3 ± 2.1; 18.3	18.3 ± 2.3; 18.3	18.4 ± 1.9; 18.3
α + β + γ	27.2 ± 2.4; 27.0	27.1 ± 2.4; 26.8	27.2 ± 2.4; 27.1

α—alpha angle—the inclination of the lumbosacral segment; β—beta angle—the inclination of the thoracic-lumbar segment; γ—gamma angle—the slope of the upper thoracic segment; x—mean; SD—standard deviation; M—median; *n*—number of participants; *p* < 0.05.

**Table 3 jcm-11-01653-t003:** Results of the rotational test (RT).

RT	All *n* = 189	Girls *n* = 85	Boys *n* = 104
x ± SD (pts)	6.5 ± 2.9	6.2 ± 2.6	6.7 ± 3.2
M (pts)	6.0	6.0	6.0
(min/max)	1/14	2/12	1/14

RT—rotational test; pts—points; M—median; *n*—number of participants; min/max—minimum value/maximum value.

**Table 4 jcm-11-01653-t004:** Spearman’s correlation between selected variables and RT, as well as between the α angle and bodyweight in the study group.

Variable	Rs	*p*
Age/RT	−0.13	0.07
Weight/RT	0.35	<0.001
Height/RT	0.08	0.3
α/RT	0.15	0.04
β/RT	−0.08	0.3
γ/RT	−0.09	0.2
α + β/RT	0.07	0.4
β + γ/RT	−0.10	0.15
α + β + γ/RT	−0.02	0.7
α/Weight	0.30	<0.001

RT—rotational test; α—alpha angle—the inclination of the lumbosacral segment; β—beta angle—the inclination of the thoracic-lumbar segment; γ—gamma angle—the slope of the upper thoracic segment; *p* < 0.05.

**Table 5 jcm-11-01653-t005:** Spearman’s correlation between selected variables and RT in the group of girls.

Variable	Rs	*p*
Age/RT	−0.16	0.14
Weight/RT	0.32	0.0032
α/RT	0.26	0.015
β + γ/RT	−0.13	0.2

RT—rotational test; α—alpha angle—the inclination of the lumbosacral segment; β—beta angle—the inclination of the thoracic-lumbar segment; γ—gamma angle—the slope of the upper thoracic segment; *p* < 0.05.

**Table 6 jcm-11-01653-t006:** Spearman’s correlation between selected variables and RT in the group of boys.

Variable	Rs	*p*
Age/RT	−0.12	0.2
Weight/RT	0.37	<0.001
α/RT	0.06	0.5
β + γ/RT	−0.09	0.4

RT—rotational test; α—alpha angle—the inclination of the lumbosacral segment; β—beta angle—the inclination of the thoracic-lumbar segment; γ—gamma angle—the slope of the upper thoracic segment; *p* < 0.05.

## Data Availability

The datasets used and/or analysed during the current study are available from the corresponding author on reasonable request.

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
