# Peer review of "Analysis of the Ability to Tolerate Body Balance Disturbance in Relation to Selected Changes in the Sagittal Plane of the Spine in Early School-Age Children"

_jcm, 2022, doi:10.3390/jcm11061653_

Round 1

Reviewer 1 Report

The manuscript “Analysis of the ability to tolerate the body balance disturbance in relation of selected changes in the sagittal plane of the spine in early school age children” (jcm-1608542) by Kurzeja et al. estimate the ability to tolerate the body balance disturbance in relation of selected changes in the sagittal plane of the thoracic and lumbar spine in early school age children. This manuscript is well written and systematically presented.

Author Response

Thank you very much for your time and experience in providing feedback on the manuscript as well as valuable comments and suggestions regarding the article. Below is the answer to your review.

The manuscript “Analysis of the ability to tolerate the body balance disturbance in relation of selected changes in the sagittal plane of the spine in early school age children” (jcm-1608542) by Kurzeja et al. estimate the ability to tolerate the body balance disturbance in relation of selected changes in the sagittal plane of the thoracic and lumbar spine in early school age children. This manuscript is well written and systematically presented.

Response: Thank you very much for your positive opinion about our research. We are very pleased that you rated it so highly. We have corrected the English. It was made by a native speaker. The certificate of this correction is attached to the manuscript.

Reviewer 2 Report

This is very interesting article describing the relation between body weight and posture. There are some points to be clarified to be accepted.

1) Authors included " Interview " in method part, but no data including contents of interview were not shown in results part.

2) It is unclear how to measure sagittal parameters such as alpha, beta, and gamma angle. I understood authors selected anatomical points on the body surface, however the method of calculating sagittal angle from them is unclear. This is crucial point of the research. Please describe about that. Moire methods is  useful to understand the curvature of coronal plane. Can  MORA 4th Generation system be used for calculating sagittal parameters, neither?

3) Children's posture and BBDT may be related with flexibility of the spine and other joints including hip, knee etc.. rather than muscle tonus. Is there any papers describing about them? If yes, please add to the citation. If no, please state authors' idea about that ( no need add to the discussion part). 

Author Response

Thank you very much for your time and experience in providing feedback on the manuscript and valuable comments and suggestions regarding the article. All comments were taken into account. Below is the answer for your review.

Reviewer 2
This is very interesting article describing the relation between body weight and posture. There are some points to be clarified to be accepted.
1.    Authors included " Interview " in method part, but no data including contents of interview were not shown in results part.
Response: the interview concerned data such as the child's age, which is presented in Table 1.
2.    It is unclear how to measure sagittal parameters such as alpha, beta, and gamma angle. I understood authors selected anatomical points on the body surface, however the method of calculating sagittal angle from them is unclear. This is crucial point of the research. Please describe about that. Moire methods is  useful to understand the curvature of coronal plane. Can  MORA 4th Generation system be used for calculating sagittal parameters, neither?
Response: The projection Moire technique also allows to assess the changes that occur in the posture in relation to the sagittal plane.
ALFA - inclination of the lumbosacral segment. It is the angle between the vertical line and the line between S1 (the spinous process of the first sacral vertebra) and the LL (peak of lumbar lordosis).
BETA - inclination of the thoracic-lumbar section. The angle between the vertical line and the line between LL (peak lumbar lordosis) and KP (peak thoracic kyphosis).
GAMMA - inclination of the thoracic - upper segment. The angle between the vertical line and the inclusive line KP (peak of thoracic kyphosis) and C7 (spinous process of the seventh cervical vertebra).
The photo is in file. 
Information was added to the manuscript. 

3.    Children's posture and BBDT may be related with flexibility of the spine and other joints including hip, knee etc.. rather than muscle tonus. Is there any papers describing about them? If yes, please add to the citation. If no, please state authors' idea about that ( no need add to the discussion part). 
Response: There is no doubt that the main condition for shaping the correct body posture is the correct structure of the musculoskeletal system. Thus, any abnormalities within the musculoskeletal system during posturogenesis may predispose to pathological changes related to shaping the body posture. These abnormalities may be, for example, changes in the function of the joints related to, for example, limitation of their mobility or loss of flexibility. In further studies, the authors intend to extend the analysis to the measurement of the mobility of the spine and selected joints of the lower extremities. It should be noted, however, that the ability to tolerate imbalances will depend, like balance, on the proper interaction of several senses, namely the vestibular organ, the sight, the sense of deep feeling, touch, and also the organ of hearing. In subsequent studies, the authors will consider extending the analysis with these additional components.

Reviewer 3 Report

The present article is an observational study that observed a direct correlation among weight and sagittal spinal curves in children of 7-10 years. The findings also pointed to a deterioration of body balance disturbance skills in those participants with higher lumbar lordose. All this information is supplied according to STROBE guidelines and in a correct English language. In order to improve the quality of this work, I suggest the authors to solve the following issues:

References 2 and 3 are monographs of local universities. I suggest the authors to use studies published in international journals worldwide accessible and in the last 5 years.

Line 85: the number of enrolled patients should be placed at the beginning of results section, not in methods. Please remove.

Line 89: comorbidities that might affect the result: please specify those considered

Line 102: please leave a line of space separating exclusion criteria from the parts of the research.

Line 106: please state the equipment o materials used to measure weight, height the course of the spinal processes… everything must be clear enough allow its reproduction.

Line 133: if possible, used English in the descriptions of Figure 1.

Line 144: what minimum height of jump is considered valid for an attempt? I think the higher the jump is, the more deviation could be observed. Please explain how this was controlled or if this does not need to be controlled.

Flow Diagram is named as Figure 1 although there is previously a Figure 1. Moreover, in my opinion the Flow Diagram is not clear: why are there more participants allocated to evaluation than randomized?

The footnotes of table 2 must explain what angles were named as alpha, beta, etc. as this explanation is only at the text.

Footnotes of table 3 should contain definitions for “n”, “M”, “(min/max)”.

Line 309: I think the authors wanted to say “tone” instead of “contractions”. Please check

Line 312: once again, I am not sure if the appropriate term is “contracture”. Please check

Line 320: please define “n.s”.

Line 324: I think the discussion could be richer if authors include in the analysis of body posture correction techniques such as global postural reeducation, for example:

-https://pubmed.ncbi.nlm.nih.gov/27386815/

-https://pubmed.ncbi.nlm.nih.gov/29037786/

-https://pubmed.ncbi.nlm.nih.gov/28660870/

Line 370: please cite some of those ambiguous reports.

I hope this information is of help.

Author Response

Thank you very much for your time and experience in providing feedback on the manuscript and valuable comments and suggestions regarding the article. All comments were taken into account. Below is the answer for your review.

Reviewer 3.
The present article is an observational study that observed a direct correlation among weight and sagittal spinal curves in children of 7-10 years. The findings also pointed to a deterioration of body balance disturbance skills in those participants with higher lumbar lordose. All this information is supplied according to STROBE guidelines and in a correct English language. In order to improve the quality of this work, I suggest the authors to solve the following issues:

1.    References 2 and 3 are monographs of local universities. I suggest the authors to use studies published in international journals worldwide accessible and in the last 5 years.
Response: The text has been changed in line with the new references items:
1.    Yang L, Lu X, Yan B, Huang Y. Prevalence of Incorrect Posture among Children and Adolescents: Finding from a Large Population-Based Study in China. iScience 2020;23(5):101043
2.    Dalise S, Azzollini V, Chisari C. Brain and Muscle: How Central Nervous System Disorders Can Modify the Skeletal Muscle. Diagnostics (Basel) 2020;10(12):1047. 
3.    Blakemore SJ, Choudhury S. Development of the adolescent brain: implications for executive function and social cognition. J Child Psychol Psychiatry, 2006 Mar-Apr;47(3-4):296-312.
2.    Line 85: the number of enrolled patients should be placed at the beginning of results section, not in methods. Please remove.
Response: The number of patients from Line 86 was removed. The results show an initial number of patients.
3.    Line 89: comorbidities that might affect the result: please specify those considered
Response: Added: diagnosed neurological diseases, posture defects, injuries of the musculoskeletal system (vision defects, disturbances in neuromuscular coordination, excessive body weight, past injuries of the spine and lower limbs).
4.    Line 102: please leave a line of space separating exclusion criteria from the parts of the research.
Response: a break line has been added.
5.    Line 106: please state the equipment o materials used to measure weight, height the course of the spinal processes… everything must be clear enough allow its reproduction.
Response: Weight and height were measured with a legalized medical column scale C315.60/150.OW-3 – 100-200cm height measuring device. The medical skin marking marker from Covidien was used to mark the characteristic anthropometric points on the skin.
Information was added to the manuscript.
6.    Line 133: if possible, used English in the descriptions of Figure 1.
Response: changed to the Moire System.
7.    Line 144: what minimum height of jump is considered valid for an attempt? I think the higher the jump is, the more deviation could be observed. Please explain how this was controlled or if this does not need to be controlled.
Response: Thank you very much for your suggestion. It can be assumed that sometimes a higher jump may result in a worse result, perhaps it will be similar sometimes with a too low jump. However, the methodology of this test was based 100% on the development of the creator of this test, Prof. Kalina. According to his assumptions, it is not the height of the jump that matters, but the method of landing on the line. We focused on this, and the height of the jump was controlled by the child. Based on the results of the repeatability and reliability of the test, we had no grounds to reject this methodology.
1.    Kalina RM, JagieÅ‚Å‚o W, BarczyÅ„ski B. The method to evaluate the body balance disturbation tolerance skills – validation procedure of the ‘Rotational Test’. Arch of Budo 2013:9:59-80.
8.    Flow Diagram is named as Figure 1 although there is previously a Figure 1. Moreover, in my opinion the Flow Diagram is not clear: why are there more participants allocated to evaluation than randomized?
Response: Changed to Figure 3. Sorry, there was an error. It should be 189, not 215. This has been corrected.
9.    The footnotes of table 2 must explain what angles were named as alpha, beta, etc. as this explanation is only at the text.
Response: Explanations under Table 2. have been added
10.    Footnotes of table 3 should contain definitions for “n”, “M”, “(min/max)”.
Response: Explanations under Table 3. have been added
11.    Line 309: I think the authors wanted to say “tone” instead of “contractions”. Please check
Response: It should be "contractions", not "tone". Contraction as a shortening of the muscle length.
12.    Line 312: once again, I am not sure if the appropriate term is “contracture”. Please check
Response: Contracture as a condition of shortening and hardening of muscles, tendons, or other tissue, often leading to deformity and rigidity of joints.
13.    Line 320: please define “n.s”.
 Response: Changed to „nonsignificant difference”.
14.    Line 324: I think the discussion could be richer if authors include in the analysis of body posture correction techniques such as global postural reeducation, for example:
-https://pubmed.ncbi.nlm.nih.gov/27386815/
-https://pubmed.ncbi.nlm.nih.gov/29037786/
-https://pubmed.ncbi.nlm.nih.gov/28660870/
15.    Line 370: please cite some of those ambiguous reports.
Response: Thank you very much for your valuable suggestion for the Discussion. We have attached the proposed articles to the Discussions in the context of future research. The research presented in our article is purely observational. Our assumption was to evaluate the relationship between posture control and posture defects. The next step will be to introduce a possible therapy for patients, taking into account the results of own research, and then one of the elements could be the addition of global postural reeducation exercises in to it and the evaluation of this therapy using the stabilographic platform.

Round 2

Reviewer 3 Report

I want to thank the authors for consider my suggestions for their work. Most of them have been successfully implemented.

The only suggestion that remains is that according to journal rules, any device used must be followed by (the name of the manufacturer, state (if USA) and country). Please apply this in line 117.

Moreover, I recommend the authors to improve the quality of Figure 3 (Flow diagram) as it appears pixelated.

Finally, my comment regarding Figure 1 meant that inside the picture there is text in Polish that should be changed to English. It is just a recommendation for authors, if they cannot changed the figure, maybe they should use another one or explain why this one is so important with words like "rejestrowany" or "zlacze" that are not understandable by most of the readers.

Author Response

Thank you very much for re-reviewing. Of course, all comments were included. Hope everything will be fine now.
Reviewer 3.
I want to thank the authors for consider my suggestions for their work. Most of them have been successfully implemented.
The only suggestion that remains is that according to journal rules, any device used must be followed by (the name of the manufacturer, state (if USA) and country). Please apply this in line 117.
Moreover, I recommend the authors to improve the quality of Figure 3 (Flow diagram) as it appears pixelated.
Finally, my comment regarding Figure 1 meant that inside the picture there is text in Polish that should be changed to English. It is just a recommendation for authors, if they cannot changed the figure, maybe they should use another one or explain why this one is so important with words like "rejestrowany" or "zlacze" that are not understandable by most of the readers.
Response: We have added the names of the manufacturers. We improved the quality of the Flow Diagram and translated the text on Figure 3. I am so sorry for the inaccuracies in understanding your earlier suggestions.